# Do Neural Stem Cells Have a Choice? Heterogenic Outcome of Cell Fate Acquisition in Different Injury Models

**DOI:** 10.3390/ijms20020455

**Published:** 2019-01-21

**Authors:** Felix Beyer, Iria Samper Agrelo, Patrick Küry

**Affiliations:** Department of Neurology, Medical Faculty, Heinrich-Heine-University, D-40225 Düsseldorf, Germany; felix.beyer@uni-duesseldorf.de (F.B.); iria.samperagrelo@med.uni-duesseldorf.de (I.S.A.)

**Keywords:** neural stem cell, subventricular zone, subgranular zone, CNS injury, disease, regeneration, transplantation, therapy, injury environment, regional heterogeneity

## Abstract

The adult mammalian central nervous system (CNS) is generally considered as repair restricted organ with limited capacities to regenerate lost cells and to successfully integrate them into damaged nerve tracts. Despite the presence of endogenous immature cell types that can be activated upon injury or in disease cell replacement generally remains insufficient, undirected, or lost cell types are not properly generated. This limitation also accounts for the myelin repair capacity that still constitutes the default regenerative activity at least in inflammatory demyelinating conditions. Ever since the discovery of endogenous neural stem cells (NSCs) residing within specific niches of the adult brain, as well as the description of procedures to either isolate and propagate or artificially induce NSCs from various origins ex vivo, the field has been rejuvenated. Various sources of NSCs have been investigated and applied in current neuropathological paradigms aiming at the replacement of lost cells and the restoration of functionality based on successful integration. Whereas directing and supporting stem cells residing in brain niches constitutes one possible approach many investigations addressed their potential upon transplantation. Given the heterogeneity of these studies related to the nature of grafted cells, the local CNS environment, and applied implantation procedures we here set out to review and compare their applied protocols in order to evaluate rate-limiting parameters. Based on our compilation, we conclude that in healthy CNS tissue region specific cues dominate cell fate decisions. However, although increasing evidence points to the capacity of transplanted NSCs to reflect the regenerative need of an injury environment, a still heterogenic picture emerges when analyzing transplantation outcomes in injury or disease models. These are likely due to methodological differences despite preserved injury environments. Based on this meta-analysis, we suggest future NSC transplantation experiments to be conducted in a more comparable way to previous studies and that subsequent analyses must emphasize regional heterogeneity such as accounting for differences in gray versus white matter.

## 1. Introduction

Ever since the discovery of naturally occurring neural stem cells (NSCs) residing in discrete niches of the adult mammalian central nervous system (CNS) [1,2,3,4,5], these cryptic cell populations received considerable interest in terms of their contribution to brain plasticity, learning, and repair. In this regard, most work addressed structure, function, and maintenance on stem cell niches located in the subventricular zone (SVZ) of the lateral brain ventricles as well as in the subgranular zone (SGZ) of the dentate gyrus. Whereas cells with stem-like properties contained within the ependymal cell population of the adult spinal cord [6,7] received less attention. Years of research have brought advances in NSC mediated regeneration and also pointed particularly to NSC grafting into affected CNS tissues and tracts as a potential therapeutic choice for a variety of neuropathologies. Yet, no clinical trial has been able to successfully translate these approaches into clinical treatments. While the large degree of heterogeneity of applied NSCs, even when isolated from defined stem cell niches [8,9], is likely to affect reproducibility, standardization, and clinical translation, different brain regions and injury types additionally contribute to the number of parameters affecting cell fate acquisition. Most NSC mediated regeneration studies focus on stem cell modulation, induced lineage heterogeneity, and their impact on the treated injury. However, an inverse view has rarely been considered so far and is therefore the main scope of this review. In order to interpret the power of an injury microenvironment on grafted cells, one has to elucidate the effects mediated by different CNS regions on introduced cell survival, proliferation, migration, and fate acquisition. We will therefore first discuss injury-free NSC engraftment studies in order to compare different outcomes on the above-mentioned parameters. In the second part, additional impact arising from host tissue injuries and lesion inflicted reactions will be addressed.

While screening the publicly available literature, it became evident that there is a large degree of heterogeneity when it comes to the NSC transplantation procedure itself, related for example to age and species of donor- as well as host tissues, the question whether sorted/enriched cell populations versus mixed cell grafts were applied or concerning time-points at which host tissue and grafted cells were analyzed. Likewise, the localization and type of an injury prior to engraftment of stem cells, as well as their positioning within lesion zones additionally influence cellular integration and differentiation. It would therefore be important to define rate limiting and dominating parameters to ensure a larger degree of comparability across different investigations and to promote the development of protocols that will eventually lead to a successful clinical translation.

## 2. Injury-Free Neural Stem Cell Transplantation Studies

Clinical research depends on animal models, which mimic human disease or injury. For neuropathological studies of the CNS various animal models such as acute and chronic spinal cord injury (SCI); traumatic brain injury (TBI); inflammatory-, genetically-, or chemically induced demyelination/neurodegeneration have been used to assess the impact of NSC transplantation on either lesion amelioration or tissue regeneration. The outcome of these studies shows a surprising degree of variability in terms of cell fate acquisition, migration within the host tissue as well as the implanted cell’s potential to fully mature (relevant studies discussed in detail below and summarized in Table 1). To better compare and interpret these differential outcomes an initial assessment of cellular reactions in the healthy CNS is warranted. Whether transplanted NSCs strictly recapitulate a developmental-like program within a non-hostile environment, as described for NSCs and their progenitor descendants in the healthy adult SGZ [10], or whether regional CNS heterogeneity decides on donor cell parameters is therefore a key question.

Using NSCs isolated from the SVZ of either postnatal day (P)5 or P75 old mice for transplantation into the striatum, motor cortex, and lateral posterior thalamic nucleus, Seidenfaden and colleagues showed that transplanted cell survival is independent of the donor animal age—at least when comparing immature postnatal- to young adult brains [11]. Furthermore, they observed that when these cells were implanted into the healthy striatum of adult mice, they did not show any migratory preference for either gray matter (GM) or white matter (WM) structures. In terms of fate choice, these SVZ-derived NSCs that are rather primed to take on a neuronal fate in their in vivo niche [12], primarily adopted glial phenotypes equally split between astroglia and oligodendroglia. Similar cell fates were observed in the additional transplantation regions such as cerebral motor cortex and lateral posterior thalamic nucleus. Surprisingly, even when grafted cells were enriched for neuronal precursors by means of the neuronal marker polysialylated neuronal cell adhesion molecule (PSA-NCAM) and subsequently transplanted into the healthy striatum, the fate outcome in vivo was unchanged compared to non-sorted cells, indicating that striatal cues dominantly suppress neuronal lineage and favor glial descendants.

A related transplantation study used rats instead of mice for both donor cells as well as host animals [13]. However, “cells capable of proliferation and neurogenesis” [13] from the adult rat hippocampus were used as donor cells and most likely represent the NSC pool of the SGZ, which are even more neuronal primed as compared to SVZ-derived cells. Additionally, prior to transplantation these cells were passaged for over 1.5 years. Despite these numerous differences, fate acquisition was again mainly described as being glial (analyzed by morphology) upon transplantation back into the hippocampus and subsequent analysis in hippocampus and the adjacent corpus callosum. Furthermore, even though some degree of cell migration was described (adjacent to striatum and corpus callosum), it was considered as minor and the majority of cells remained close to the injection site. Of note, transplanted cells found in the corpus callosum, a region absent of neuronal cell bodies and primarily characterized by myelinating oligodendrocytes (OLs) and oligodendroglial precursor cells (OPCs), adopted an oligodendroglial morphology. Only transplanted cells directly located in the granule cell layer of the hippocampus, indeed acquired a neuronal cell fate, which, even back then, evoked the author’s statement that “the in vivo fate of these cells is clearly influenced by exogenous factors” [13]. In contrast to these two allotransplantations (transplantation in which the donor material is derived from a different donor of the same species), Raedt and colleagues xenografted (donor and host differ in species) SVZ-derived NSCs from young mice, which additionally were propagated over 10 passages prior to transplantation, into the adult rat hippocampus [14]. Similar to Gage´s observations, transplanted cells did not show high degrees of migration except for a few cells entering the granule cell layer of the hippocampus. However, in contrast to SGZ-derived NSCs, some of these SVZ-NSCs were detected close to the SVZ of the host rat, indicating SVZ niche specific cues, which might have attracted intra-hippocampal SVZ-NSC grafts. Even though oligodendroglial differentiation was not accessed, the majority of surviving NSC descendants were positive for glial markers such as astroglial glial fibrillary acidic protein (GFAP) (38.6%) and only a few cells (5.8%) showed mature neuronal marker RNA binding protein, fox-1 homolog (C. elegans) 3 (Rbfox3 or NeuN) expression, which is in agreement with earlier observations [11,13].

Further support for the assumption that exogenous factors influence the fate of transplanted NSCs [13] resulted from an extensive transplantation study published four years later [15]. While different injection sites within young adult, non-injured mice were chosen, a defined single donor cell condition was maintained for all transplantations, allowing for a direct comparison of fate acquisition and other cell properties within different brain regions. Neural stem cell survival rates were largely comparable between striatum, cortex, and olfactory bulb indicating that no preferential survival cues are expressed and secreted in these brain regions. Hippocampus implanted cells, however, survived less well for non-disclosed reasons. Moreover, it was also stated that proliferation (or even tumorigenesis resulting from aberrant cell expansion) was not observed among transplanted NSCs—an observation that was confirmed by many follow-up studies in the field. Whether this implies the existence of mechanisms that actively restrict NSC division outside of their niches or whether this is rather a consequence of host-initiated cell differentiation remains to be shown but is promising in light of preventing the generation of brain tumors such as for example glioblastoma. Given that donor cells were acutely isolated from the adult SVZ and were therefore more primed towards neuronal differentiation subsequent analysis of fate choice in this study must be interpreted carefully, as even mentioned by the authors. Cortex- and striatum-transplanted cells attained primarily non-neuronal type-C (or astrocyte) and type-A (neuronal precursor) phenotypes. On the other hand, when these cells were transplanted into the olfactory bulb, a region receiving neuronal progenitors via the rostral migratory stream (RMS) more mature granule neurons were found to descend from transplanted NSCs [15]. Moreover, significant cell migration was observed in this situation. Of note, donor SVZ cells were isolated from a transgenic mouse line in which LacZ was expressed under the control of the neuron-specific enolase (NSE) promotor, which limits a proper analysis of fate choice due to reporter restriction to neuronal descendants. Still, some degree of astrocytic phenotype acquisition was observed indicating that the transgenic NSE promotor activity was not too specific. Oligodendroglial differentiation was not assessed in this study.

For a successful treatment of neurological conditions such as multiple sclerosis (MS) or adrenoleukodystrophies transplanted NSCs would have to distribute well within diseased brains in order to access multiple and irregularly dispersed lesions. Active migration within the brain parenchyma appears indeed to be a rare feature of implanted NSCs as for example transplantation of SVZ-derived NSCs into neighboring SVZ regions of adult healthy mice did not result in any observable migration activities [16]. Transplantation into the lateral ventricle, however, resulted in migration along the RMS up to the olfactory bulb, indicating that also artificially implemented cells are restricted to naturally occurring migration routes and cues.

Supporting evidence that brain region specific cues act across different species arose from a subsequent study in which embryonic human NSCs (huNSCs) were applied to injury-free models [17]. Here, donor cells extracted from 6.5 to 9-week-old embryonic human forebrains and subsequently expanded over 9 to 21 passages were transplanted into either the dentate gyrus, the RMS, the striatum, or the SVZ of adult immunosuppressed rats. These multipotent NSCs showed no tumor formation within the first six weeks post transplantation. In line with the previous reports on rodent-to-rodent transplantations, only minor NSC migration was observed into dentate gyrus or striatum. Moreover, such low migration rates were questioned in terms of whether they result from true (host evoked) cell movement or rather from random dispersion as consequence of pressure implemented during the implantation procedure. Similar to rodent-to-rodent NSC engraftments, larger migration activities were only observed when huNSCs were transplanted into the SVZ or RMS. In there, these embryonic cells adopted features similar to those of the surrounding endogenous neuroblasts. Interestingly, despite the multipotency and the rather high proliferative capacity of these huNSCs of embryonic origin, they adopted exclusively a neuronal fate when reaching the olfactory bulb or within the neuronal SGZ of the hippocampus. On the other hand, within the striatum, both glial- and neuronal NSC derivatives were described. Since NeuN expression was still absent these were most likely immature neuronal cells—a notion which supports the above-mentioned type-A cell (neuronal precursor) generation by rodent NSCs in the striatum [15]. Absent NeuN expression was also reported for the majority of host neurons in the striatum [17]. In this regard, the use of additional neuronal markers in order to achieve a more detailed description of the acquired cell fate resulted in the observation that the acquired neuronal phenotypes of engrafted NSC derivatives matched the neuronal population of the host striatum. Here, the authors describe that the transplanted NSCs differentiated into either glutamic acid decarboxylase (GAD) 67-, calbindin-, or dopamine- and cAMP-regulated neuronal phosphoprotein- (DARPP-) 32-positive neurons, three neuronal types located in the host tissue [17].

The astonishing impact especially the RMS exerts on transplanted NSCs was corroborated by yet another study [18]. This investigation provides additional support for the SVZ/RMS environment acting as key guidance structure for transplanted cells since neonatal SVZ-derived NSCs transplanted into the neonatal SVZ migrated with the same properties (route, speed, morphology) along the RMS as the endogenous cells arising from the SVZ. Even at their final destination in the olfactory bulb, the ratio of granule- versus glomerular-cell layer infiltration was maintained at 3:1, independent of whether these were transplanted NSCs or endogenous NSC-derived neuroblasts. However, when embryonic ventricular zone-derived cells were used they found that these NSCs never entered the RMS pathway but strictly remained at the injection sites within the neonatal SVZ. The authors thus suggested that this might be due to their inability to recognize specific guidance cues along postnatal migratory routes, which could be attributed to naturally different migratory properties and their role in populating the developing neocortex [18,19]. Nevertheless, this described inability to migrate along the RMS is contradictory to results by Fricker and colleagues, who also used embryonic NSCs (although from human embryonic tissue) and reported migration along the host RMS. Apart from species-specific migratory properties, observed differences in the migratory behavior could also derive from the fact that Brock and colleagues analyzed host brains only up to 15 days post-transplantation whereas the other team analyzed host brains six weeks post-transplantation into the SVZ. Moreover, human embryonic NSCs were passaged between 9 and 12 times prior to transplantation [17], potentially impacting a number of cellular parameters (for comparison see Table 1).

## 3. Brain Pathology Models and Their Heterogenic Impact on NSC Fate

Comparing outcomes of NSC transplantation in different neuropathological or injury inflicted models is important in order to understand the adaptability NSCs are capable of and also for the development of CNS repair strategies. Nevertheless, due to the high degree of variation and heterogeneity across the different model systems this section will focus on selected models representing both common and rare as well as global and focal pathologies. Working out the details between different injury models, and taking into account heterogenic responses of different brain regions depends on comparable starting conditions. This includes information on donor age, on the particular donor stem cell niche, on isolated cell types and whether they were propagated in culture or whether genetic manipulation was applied prior to transplantation. Moreover, variations deriving from different model systems will also be considered.

### 3.1. Dysmyelinating Neuropathologies

Treatment of hereditary white matter disorders characterized by abnormal or complete absence of myelin due to mutations in genes encoding for myelin proteins (such as in Pelizaeus-Merzbacher disease) will most likely depend on engraftment of healthy cells giving rise to functionally unimpaired myelin forming oligodendrocytes. In shiverer mice (shi) homozygous mutations in the myelin basic protein (MBP) gene lead to the absence of MBP expression and consequently to low levels of compact and functional myelin [20,21] and it therefore serves as an animal model for dysmyelinating neuropathologies. Intracerebroventricular transplantation of C17.2 NSCs (an immortalized cell line derived from neonatal mouse cerebellum) into newborn (P0) shi mice resulted in excessive cell migration within the brain parenchyma, both into GM and WM. The degree of oligodendroglial differentiation was significantly higher when engrafted into these myelination deficient mice as compared to wildtype hosts. While in healthy mice 16% of the transplanted cells differentiated into the oligodendroglial lineage, up to 28% of them generated oligodendroglia in the shi brain [22]. This might therefore reflect the need for myelinating glial cells in the shiverer CNS but also the capacity of NSCs to sense and react to such a deficient background. This is a promising hint that NSCs can indeed compensate according to missing/impaired cell types and, as stated by the authors: “Such behavior might reflect a fundamental developmental strategy with therapeutic utility”.

However, intracerebroventricular transplantation of the same NSC cell line (C17.2) into shi mice of comparable age (P1-P3) resulted in mainly neuronal differentiation [23]. While this was primarily a feasibility study and the primary focus of this study was not on cell fate acquisition, the authors still state that no oligodendroglial differentiation, and hence no myelin production, was observed. Transplanted NSCs did accumulate into the surrounding parenchyma and expressed the neuronal marker TuJ-1 [23]. Interestingly, C17.2 cells were a courtesy of the former-mentioned study´s senior author (Prof. Evan Snyder) and in both studies homozygous mouse mutants (*shi/shi*) were used as host animals, which in terms of methodology makes these studies highly comparable. Therefore, the reason for such differential fate outcome remains obscure.

Studies in which transplantation experiments were performed using different (neural) stem cell types but exclusively applying a single neuropathological model are of special value since methodological differences can be ruled out. When adult forebrain SVZ neural precursor cells (aNPCs) were grafted into the dysmyelinated spinal cord of shiverer rats, the majority differentiated into oligodendroglial cells and even into mature myelinating OLs [24]. In contrast, transplantation of embryonic stem cell-derived primitive neural stem cells (pNSCs) into the same model did not result in successful integration and even led to heterotoma formation. However, definitive NSCs (dNSCs) which were also investigated in this study and which derived from pNSCs by leukemia inhibitory factor (LIF) and fibroblast growth factor 2 (FGF2) application revealed oligodendroglial fate acquisition and generation of mature myelinating cells comparable to aNPC descendants. For dNSCs and aNPCs differentiation into oligodendroglia was in the range of 48–58% whereas only 3–4% astrocytes and 3–4% neurons were observed. Thus, a natural restriction point for (neural) stem cells appears to exist deciding on whether and when they are able to properly react to neuropathological dysfunctions and deficits, at least in a dysmyelinated environment such as the shiverer rodent.

### 3.2. Traumatic Brain Injury

Traumatic brain injury (TBI) results from forced impact to the skull and brain, which leads to a primary- (direct negative influence on tissue architecture and homeostasis) as well as a secondary (cell death, inflammation) injury [25]. Due to their multipotent nature, NSC transplantation into TBI lesions is thus considered as a promising approach for broad cell replacement and functional improvement.

The study by Koutsoudaki and colleagues elegantly demonstrated the non-necessity of NSC modulation prior to transplantation into injured adult mouse brains. Upon TBI to the hippocampus by stabbing multiple times 2 mm deep through the cortex, corpus callosum, and hippocampus, non-modified (reporter-gene expression only) and insulin like growth factor 1 (IGF1) overexpressing NSCs (mouse P5, SVZ-derived) were transplanted close to the lesion site. Considering fate outcome as well as functional improvement (as assessed by the Morris Water Maze test, MWM), no differences between IGF1 producing and reporter gene only expressing NSCs were observed. Both cell populations ameliorated injury-induced spatial learning deficits and in both cases transplanted NSCs mainly differentiated into oligodendroglial cells [26]. As in some hippocampal injury procedures, also white matter structures of the corpus callosum are disrupted resulting in focal oligodendrocyte death [27], subsequent migration of NSCs into the corpus callosum and oligodendroglial differentiation (both in the corpus callosum and hippocampus) are probably driven by the need to repair WM structures. However, this observation is somehow in contrast to what has been described in injury-free hippocampus transplantations, where mainly astroglial and neuronal differentiation was described [13,14]. Since in both experimental set-ups (studies by Gage et al. and Raedt et al.) fate acquisition was largely comparable despite different donor cell origin, differences in terms of a higher OL differentiation in Koutsoudaki´s TBI model might be best explained by an injury-dependent change in the tissue microenvironment.

In contrast to the TBI wound induced by a blunt-end needle, which also disrupts WM structures [26], the modified Feeney method was used to injure the cortical motor area in rats [28]. Unfortunately, from the data presented in this study it is not clear whether this modified method also results in hemorrhage in the underlying WM leading to the formation of a necrotic cavity within the corpus callosum as originally described [29]. According to the authors, transplantation of mouse embryonic hippocampus-derived NSCs 3 mm from the lesion (rostral, caudal, left, and right) at the day of injury resulted in migration of these NSCs towards the injured region. Analysis of fate choice revealed that a few transplanted NSCs gave rise to TuJ-1-positive neurons and GFAP-positive astrocytes [28]. Differentiation into oligodendroglial cells was not reported so it can only be assumed that this modified procedure did not harm WM structures.

In a comparable experimental set up (modified Feeney method) neonatal mouse hippocampus-derived NSCs were transplanted 24 h after injury into the pericontusional region [30]. Similar to the above-mentioned investigation transplanted NSCs survived, migrated towards the lesion site differentiating into few TuJ-1-positive neurons and GFAP-positive astrocytes [30]. Although this study focused on BDNF expression-mediated effects of NSCs on the extent of injury and subsequent functional improvements, a proper description of all three cell types (neuronal, astroglial, as well as oligodendroglial cells) was not provided.

### 3.3. Temporal Lobe Epilepsy

Temporal lobe epilepsy (TLE) is a common form seen in about 30% of epileptic patients. By injecting kainic acid (KA) into the rodent hippocampus or the SVZ, neurodegeneration is induced and neuronal cell death and functional impairments similar to TLE patients can be mimicked. While NSC transplantation has also been investigated in terms of trophic support these NSCs can confer to the damaged environment, this section will focus on the aspect of NSC mediated cell replacement and fate choice.

Since kainate induced hippocampal degeneration represents a focal CNS injury, transplantation of P5 mouse SVZ-derived NSCs into the hippocampus close to the injection site resulted in only minor migration towards the lesion area [31]. Stem cell marker (nestin) expression as well as transplanted NSC proliferation were rarely or not all observed after longer time-points (up to 60 days) post-transplantation [31]. Interestingly, transplantation of NSCs directly into such kainate-treated tissues led to significant differences in fate choice compared to non-injured or TBI-injured hippocampi. Basically no glial cells as assessed by GFAP- and O4 staining for astrocytes and oligodendroglia, respectively, derived from the transplanted NSCs within the kainate injured hippocampus. Instead, approximately 36% of the engrafted cells were immunopositive for the neuronal marker NeuN and the majority of cells expressed the neuronal progenitor marker doublecortin (Dcx) [31]. Further highlighting the major impact an injury microenvironment can exert on engrafted NSCs, the authors also demonstrated that only few or even no beneficial effects could be attributed to IGF1 overexpressing NSCs when looking at later time-points. In fact they observed, that the naïve NSCs did not express IGF1 in culture but rather adapted this beneficial feature upon transplantation into the kainate-treated hippocampus. The exact reason for the cognitive improvement (MWM test) upon NSC transplantation needs further investigation since several beneficial effects of the NSC transplants were described such as neuronal cell replacement, IGF1 expression, decreased astroglial activation as well as normalized proliferation rates in the dentate gyrus. In terms of clinical translation, these results might also be promising for the development of potential treatment options for patients with TLE.

While transplantation of NSCs into the kainate-induced neurodegenerative hippocampus resulted in mostly neuronal differentiation and ameliorated cognitive function [31], another TLE study using NSCs derived from the embryonic rat medial ganglionic eminence (mgeNSCs) showed substantial different cell fates and behavioral outcome. Hippocampal neurodegeneration was induced by injecting kainate intraperitoneally in rats to generate a chronic injury and NSCs were therefore transplanted several months later [32]. In this situation, the majority of transplanted mgeNSCs differentiated into astrocytes (57%). Only a minor fraction differentiated into mature NeuN expressing neurons (13%), GABAergic interneurons (10%) and into few OPCs (3%). This finding seems contradictory to the hypothesis that the same kind of injury microenvironment exerts a dominant impact on the fate of (different) engrafted NSCs, indicating that timing is another critical factor, not only in terms of survival and integration as revealed for spinal cord injuries but also for acquired cell fates. It can therefore be assumed that chronic neurodegeneration has changed the injury microenvironment so that it mainly contains astrogenic cues (perhaps in a more rapid manner than the healthy hippocampus as it becomes more astrogenic with age anyways [33]). This is also supported by the observation that the majority of cultured mgeNSCs differentiated into neurons and OPCs in vitro [32]. Consequently, while in acute TLE lesions neuronal fate acquisition and cognitive improvement were observed [31], NSC transplantation in chronic TLE lesions resulted in astroglial differentiation, subsequent GDNF expression, a restored GDNF expression in host astrocytes, no cognitive improvement (MWM) but beneficial effects on spontaneous recurrent motor seizures (SRMS) [32]. How engrafted NSCs sense such subtle environmental changes remains to be investigated. Of note, no or only minor migration of transplanted cells within the hippocampal tissue and no tumor formation was observed in both acute and chronic TLE models.

A further TLE study also reported primarily astroglial fate choices although NSCs were transplanted into an acute TLE model [14]. A change in the injury microenvironment over time was supported by the observation that transplantation of mouse SVZ neurospheres at three weeks post-kainate lesion significantly improved the survival of transplanted cells compared to implantation after only three days. The striking differences in fate outcome despite similar starting conditions might indeed be dependent on subtle variations of transplanted cells. Raedt and colleagues used NSCs derived from young adult mouse SVZ, which were passaged at least ten times in vitro prior to transplantation, whereas Miltiadous’ team used NSCs derived from P5 mice, which were propagated for “at least 3–4 passages”. Moreover, while Raedt and colleagues xenografted mouse cells into the rat TLE model, the other group induced TLE in mouse hippocampi and grafted mouse cells. In addition, while Raedt’s team transplanted SVZ NSC-derived neurospheres into kainate injection sites, Miltiadous’ group transplanted dissociated SVZ-derived NSCs 600 µm away from the initial kainate injection site into the more rostral part of the injured hippocampus. Finally, the xenografting procedure was accompanied by cyclosporine application for immunosuppression, which can exert additional influences on the injury microenvironment due to a diminished immune response. Of note, beneficial effects of cyclosporine have been described for different injury models [34,35,36].

Another interesting finding is that SVZ-derived NSCs transplanted into healthy non-injured hippocampi were also located close to the SVZ several weeks after transplantation [14]—a distribution that was, however, not seen in TLE animals. This suggests that SVZ-derived NSCs can sense cues from the SVZ, which can be overridden by signals from the injured hippocampal tissue. Such a potential signal hierarchy with injury-derived signals dominating over naïve cues could be promising when it comes to NSC transplants in neuropathologies featuring dispersed lesions such as in MS.

### 3.4. Sly Disease

Sly disease is a rare hereditary lysosomal storage disorder, characterized by the deficiency in β-glucuronidase (GUSB) and subsequent accumulation of glycosaminglycans in many organs including the brain leading to mental retardation. Applying NSCs as early as technically feasible might provide a potential therapeutic approach in order to restore global dysfunction of the developing brain. The mucopolysaccharidosis VII (MPS VII) mouse strain mimics human sly disease pathological features. Upon transplantation of GUSB-expressing C17.2 NSCs into lateral ventricles of neonatal MPS VII mice, cells distributed and integrated in the whole brain and no tumorigenesis was observed [37]. Widespread integration into most brain regions (from olfactory bulb back to the hippocampal area) is most likely attributed to the global impact of the developing brain since a similar distribution was described in healthy animals [37]. Interestingly, donor cells did not spread completely throughout the whole brain, since in regions to which cells from the host SVZ do not contribute (e.g., retina) GUSB activity from donor cells was absent. Integration into the host brain tissue was maintained for up to 12 months post-transplantation indicating that even in a less beneficial environment (MPS VII brain) NSCs might be capable of surviving and contributing to normal CNS homeostasis. Morphological analysis three weeks post-transplantation revealed “normal neural morphologies” [37]. However, a detailed immunohistochemical characterization was missing. In light of the observations from this study it will be of interest to see whether also in mouse injury models with distinct focal brain region impairment (e.g., stab wound to the cortical tissue) NSC transplantation into the neonatal developing brain would result into more widespread cell integration as opposed to the generally observed focal NSC occurrence around adult lesion sites.

### 3.5. Stroke

The lack of oxygen supply in the brain results in ischemic stroke, leading to irreversible neuronal damage [38]. Ischemia can be induced in the rodent brain via middle cerebral artery occlusion (MCAO) resulting in a similar striatal injury as in stroke patients. Upon intrastriatal transplantation of embryonic cortical mouse NSCs, which in vitro primarily express nestin and immature neuronal markers (Dcx, β-III-Tub) and incorporate BrdU (labeling proliferative cells), into non-MCAO rat brains, these cells did not migrate and the majority died within a few days [39]. However, in MCAO rats where the ischemic epicenter lies in the striatum, transplanted cells survived and migrated throughout the ischemic striatum [39]. In terms of fate acquisition, sham and MCAO striatal tissue both allowed differentiation of engrafted NSCs into neuronal cells, astrocytes and oligodendrocytes. The majority of NSCs in fact differentiated into GFAP-positive astrocytes and into Dcx-positive neuronal precursors—a pattern that was also described upon grafting of embryonic human NSCs into the healthy striatum [17]. Of note, the number of generated mature NeuN-positive neurons increased in the MCAO group compared to sham animals six days post-transplantation [39]. Interestingly, upon transplantation into the lateral ventricle of the adult rat (MCAO or sham), robust migration into the striatum was only observed in the MCAO group [39]. Here, assessment of fate choice in the ischemic striatum revealed an increase in Hu-positive neuronal cells while astroglial differentiation was decreased compared to sham operated animals six days post-intracerebroventricular transplantation. The high proportion in astroglial descendants of the intrastriatal—compared to intracerebroventricular—transplantations could thus reflect the acute necessity for functional astrocytes following MCAO. Furthermore, while transplantation of neonatal mouse SVZ-derived NSCs into the rat striatum 48 h following MCAO also resulted in neuronal differentiation (22%) accompanied by cells with astroglial- and oligodendroglial fates, co-transplantation of astrocytes and NSCs resulted in an increase in generated neurons (37%) [40]. Such co-transplantation also led to a higher ratio of proliferating and surviving NSCs seven days after transplantation.

Transplantation of neurosphere-derived cells from neonatal mice hence containing NSCs, various progenitors and more differentiated cells [41] into the ventricle of 12-week-old mice four hours following MCAO resulted in migration into striatal- and cortical tissue. Sham operated animals receiving the same grafts were devoid of donor cells in the brain parenchyma [41]. Even though neuronal fate acquisition was not assessed, early time-point analyses (one and seven days post-transplantation) revealed nestin-, GFAP-, as well as chondroitin sulfate proteoglycan 4- (Cspg4 or NG2) positive NSC descendants which, however, had disappeared after 14 days. Assessment of mRNA expression in the cortex revealed an increase in various cytokine- and trophic factor messages (such as CXCL12, TGF-β1, VEGF-A, IGF1, and BDNF) in the transplantation-free MCAO-group compared to sham operated animals, which points towards a rapid host initiated regeneration reaction. Interestingly, the authors report further increased transcript levels in the ischemic group, which received NSC transplants [41]. Even though mRNA origin (host or donor) could not be discriminated, transplanted NSCs thus appear to also modulate endogenous repair mechanisms.

Regarding differences in the microenvironment of ischemic brain regions over time, Darsalia and colleagues compared the differential outcome of intrastriatally transplanted human fetal striatal NSCs 48 h and six weeks after stroke. Transplantation after 48 h following stroke resulted in a higher NSC survival rate as compared to transplantation at the later time-point. However, different time-points did not affect the extent of migration, differentiation and proliferation. Interestingly, analysis of neuronal fate acquisition revealed no change in the percentage of Dcx-positive neuroblasts but for both transplantation time-points the percentage was lower as compared to injury-free animals [42]. Such observations are indeed relevant in light of the necessity to conduct autologous transplantation procedures at rather late stages in stroke patients. Moreover, looking at the currently available studies it becomes clear that fate acquisition of transplanted NSCs in the MCAO stroke model must be analyzed more carefully in future studies taking into account all neural lineages NSCs can give rise to, their maturation kinetics as well as subregional differences. This is even more important in light of the current assumption that transplanted NSCs can reflect, to some degree, the regenerative need of an injury environment.

### 3.6. Multiple Sclerosis

Multiple sclerosis (MS) is an autoimmune disease characterized by oligodendrocyte and myelin loss, which ultimately leads to axonal degeneration and subsequent sensory, motor, and cognitive dysfunction [43]. The most common animal model used for recapitulation of mainly inflammatory aspects of this disease is experimental autoimmune encephalomyelitis (EAE). The majority of the here considered studies used myelin oligodendrocyte glycoprotein (MOG_35–55_) peptide induced EAE, which leads to a T-cell mediated autoimmune response towards oligodendrocytes with the first symptoms appearing ten days after initial immunization.

Intracerebroventricular transplantation of human embryonic stem cell-derived early multipotent neural precursor cells (hESC-NPCs) demonstrated robust migration into the host WM, which led to significantly reduced clinical sings of EAE in host mice [44]. The beneficial effects presumably resulted from diminution of inflammatory processes and subsequent reduced demyelination, as well as axonal damage [44].

Further evidence for a high migratory activity of implanted cells in the EAE brain derived from a report on human induced pluripotent stem cell-derived NSCs (iPSC-NSCs) having migrated into the dentate gyrus one month following intraventricular transplantation in EAE mice [45]. Co-localization of reporter protein-positive iPSC-NSCs and neuronal protein TuJ-1 indicated integration of graft-derived neurons into injured areas of the dentate gyrus. Two months following transplantation endogenous remyelination in the marginal zone of the WM was detectable. Thus, iPSC-NSCs dramatically reduced T-cell infiltration and ameliorated EAE dependent demyelination resulting in functional recovery [45]. It seems striking that in contrast to dysmyelinating pathologies [22,24] no substantial oligodendroglial differentiation of transplanted NSCs was described. Since endogenous remyelination was in fact detected, such effects either resulted from an immunomodulatory action of the stem cells, along with some of them giving rise to neurons, eventually tuning down the T-cell response. Alternatively, transplanted NSCs might also have activated host OPCs to differentiate and myelinate.

Minor regeneration promoting effects using allogenic NSCs were also reported for a chronic EAE model, into which transplantation was performed 40 days post-immunization. Here, treatment with non-modified bone marrow-derived (BM) NSCs (derived from six- to eight-week-old mice) [46] blocked the demyelination process but did not favor remyelination in the diseased spinal cord. On the other hand, BM-NSCs molecularly engineered to produce LINGO-1-Fc, a soluble LINGO-1 (leucine rich repeat and IG domain containing 1) antagonist (LINGO-1´s role in neuropathologies is reviewed in [47]), significantly promoted remyelination in the chronic stage of EAE and reduced further demyelination processes as compared to control animals [48]. Additionally, LINGO-1-Fc expressing BM-NSCs significantly promoted neurological recovery. Histological assessment revealed improvements in axonal integrity, enhancement of oligodendrocyte maturation, and neuronal repopulation of the degenerated areas.

While the two previous studies report minor regeneration promoting effects using allogenic or naïve NSCs in EAE mice, Einstein and colleagues could show that intraventricularly transplanted striatal newborn rat multipotential NSCs migrated into inflamed WM tracts and subsequently differentiated into glial cells in EAE rats. This transplantation strategy resulted in a reduced inflammation of the host brain and ameliorated the disease course. Whether beneficial effects were due to additional processes attributed to particularly the rat system remains open.

Intraventricular transplantation of both, glial-derived neurotrophic factor (GDNF) overexpressing- (GDNF-NSCs) or naïve newborn rat cerebrum-derived NSCs ten days following EAE induction led to a delayed disease onset and significant reduction of clinical EAE signs [49]. More specifically, GDNF-NSC receiving rats recovered five days earlier to basal gait as opposed to rats that received non-modified NSCs. This observation might be attributed to the significant reduction in the number of inflammatory infiltrates in the striatum and the lower number of cells within each infiltrate in animals receiving naive NSCs or even to a greater extent upon application of GDNF-NSCs. Interestingly, the majority of NSCs migrated specifically towards inflamed areas in the corpus callosum and striatum. In addition, assessment of GDNF-NSC differentiation in the striatum revealed a significantly higher neuronal and oligodendroglial fate acquisition as compared to naïve NSCs. The latter of which preferentially generated GFAP-positive astrocytes.

### 3.7. Alzheimer´s Disease

Neurodegeneration in Alzheimer´s disease (AD) results in learning deficits and dementia. In a murine AD model (APP/PS1 mice), neuron-specific Thy1 promotor driving the co-expression of KM670/671NL mutated amyloid precursor protein (APP) and of L166P mutated presenilin-1 (PS1) leads to human amyloid depositions and local neuronal loss in the dentate gyrus [49,50]. The majority of hippocampus transplanted neonatal mouse SVZ-derived NSCs (primed towards the neuronal lineage by retinoic acid application in vitro) differentiated into neurons [51]. However, despite neuronal priming 24% of the total NSC population still differentiated into glial cells, implying the additional necessity to replace glial cells as well. In total, 8% NSC-derived oligodendroglia were reported—a surprising finding considering that upon NSC transplantation into the healthy hippocampus or hippocampal TLE models almost no oligodendroglial differentiation was described [14,31,32]. Unfortunately, efforts aiming at the identification of additional progenitor populations were not undertaken. A variety of AD mouse models have been generated, such as for example APP/PS1, B6C3-Tg, 5xFAD, 3xTg, APPSw-NSE (www.alzforum.org) [50,52,53,54,55] of which the APP/PS1 model is most widely used featuring amyloid plaque generation, mild to robust neuronal cell loss and cognitive impairments [51,56,57,58,59,60,61,62] and reporting on cognitive improvement after NSC transplantation [56,57,61,62]. However, Marsh and colleagues showed that fetal-derived human NSCs engrafted into the hippocampus migrated up to 1.7 mm and were detected in the lateral ventricle five months after transplantation, where they failed to differentiate and formed ectopic human cell clusters. Probably due to the lack of differentiation, the authors found no evidence for cognitive improvement [63]. Using an APPSw-NSE transgenic mouse line, lateral ventricle transplanted fetal human telencephalon NSCs (13 weeks of gestation) showed extensive migratory activity as indicated by broad distribution of these cells in SVZ, WM tracts, striatum, thalamus, hypothalamus and cortex. Some of the transplanted NSCs differentiated into neuronal- (5.8%) and glial cells (oligodendroglia 2.3% and astroglia 11.7%) but most of them remained nestin-positive (82.4%). Despite the low differentiation rate, transplantation of these NSCs resulted in improved spatial memory, decreased tau phosphorylation, lowered Aβ42 levels, and attenuated microgliosis and astrogliosis [58]. The observed contribution to regeneration was therefore unlikely a consequence of direct cell replacement but rather due to trophic, modulatory effects.

Interestingly, most of the compiled studies report that transplanted NSCs showed high migratory behavior leaving injection sites [56,57,58,59,60,61,62,63,64,65]. In addition, transplanted cells that remained mostly undifferentiated were mainly reported in cases where human NSCs were used [57,58,63]. Whether this reflects a natural restriction of human cells to adapt to an AD related environment remains to be shown. On the other hand, murine NSCs preferentially acquired either astrocytic [59,60,62,64] or neuronal fates [51]. While some studies report increased synaptic densities following NSC transplantation with a concomitant reduction in Aß concentration [56,58] others declare positive effects of transplanted NSCs without Aß alterations [51,57,62].

Using a chemical model to mimic AD related neuropathological features by okadaic acid injection into the lateral ventricles [65] only transplantation of rat NSCs (both derived from the SVZ or SGZ of embryonic rats) overexpressing human nerve growth factor (NGF-NSCs) led to robust survival, migration, and integration while non-overexpressing NSCs did not. Besides, NGF expressing cells also enhanced cognitive performance. Whether this is due to an inhibited differentiation process, as reported for other AD studies, remains to be shown as no detailed assessment of fate choice was performed [65].

### 3.8. Huntington´s Disease

Huntington´s disease (HD) is an inherited neurodegenerative disease caused by the progressive loss of GABAergic medium spiny neurons (MSNs) in the striatum. Injection of 3-nitropropionic acid (3-NP) into the striatum serves as an animal model to mimic HD symptoms and related neurodegenerative aspects. Transplantation of human NSCs (v-myc immortalized and derived from fetal telencephalic tissue) into the injured striatum one week prior to the injury resulted in decreased loss of striatal neurons as well as the appearance of calbindin-expressing (marker for medium spiny striatal projection neurons) donor cell-derived neurons [66]. Upon immunohistochemical analysis of the grafting site, the authors stated that the transplanted NSCs were able to read signals operating in the damaged striata and to appropriately differentiate into GABAergic neurons. However, transplantation of the same NSC pool 12 h after the 3-NP injection did not confer such beneficial effects. As only a single time-point at one week post-3-NP injection was immunohistochemically analyzed, it is therefore possible, that the observation window was too small in order to detect a similar cell replacement role of these NSCs in this second set-up. A further study using YAC128 mice as yet another HD animal model confirmed that transplanted mouse iPSC-derived NSCs are capable of replacing lost neurons in the striatum [67]. YAC128 mice carry a full-length human mutant huntingtin gene (mHTT), which leads to selective, age-dependent progressive impairments in motor- and cognitive functions due to neuronal loss in the striatum [68,69,70]. Upon transplantation of mouse iPSC-derived NSCs into YAC128 mice, these mice showed better motor function (rotarod test), which was accompanied by neuronal differentiation of transplanted NSCs [67]. These mature NeuN-positive neurons were also positive for DARPP-32 (medium-sized spiny striatal projection neurons)—a feature that was also observed in injury-free striatal transplantation studies [17]. Moreover, it was of interest to see that NSC transplantation into control (wildtype) mice resulted in decreased survival of the engrafted cell population as opposed to the YAC128 striatum indicating a beneficial role of the diseased host environment. This observation finds further support by a transplantation study in which another chemical HD animal model was investigated. Here, injection of quinolinic acid into the striatum leads to regional excitotoxicity and subsequent degeneration of DARPP-32-positive, medium spiny projection neurons [71,72,73]. Even though the authors referred to the implanted cells as neural precursors, corresponding in vitro analyses revealed that the isolated cells were proliferating and could give rise to both, neurons and astroglia, which suggests that they were still NSCs. Finally, transplantation of human fetal, striatal eminence-derived NSCs six hours after striatal injury induction led to broad migration rostral and caudal to the injection site and to robust tissue integration [73]. Analysis after 12 weeks post-transplantation revealed that the majority of NSCs indeed differentiated into neurons of which some also displayed DARPP-32 expression. Glial descendants were not investigated.

## 4. Heterogeneity among Spinal Cord Injury Models and Donor Cell Origin

Spinal cord injury (SCI) is a devastating neurological condition, which is caused by a traumatic impact to the spinal cord. It results in permanent impairment of motor- and sensory functions due to the interruption of descending and ascending nerve fiber tracts. Local cell loss at sites of injury is followed by glial scar formation and accompanied by inflammation which prevent regeneration of transected axons and exert an additional negative impact on functionality and survival of remote neurons [74,75]. Stem cell transplantation seems to be a beneficial therapeutic approach in order to promote spinal cord regeneration either via secretion of neurotrophic factors or in that engrafted stem cells adopt neuronal and glial identities and functionally integrate into damaged neuronal circuits [76,77]. However, the question to what extent regeneration can take place and how the lesion environment affects fate and differentiation of transplanted NSCs is still under investigation addressing different spinal cord injury models as well as NSC populations of different origin.

In this regard, we compared data on fate acquisition of transplanted NSCs from two methodologically different SCI models. Compression or contusion used to induce broader and probably medically more relevant spinal cord lesions is compared to more defined injuries resulting from spinal cord hemisection or complete transection. Both models affect spinal cord integrity and lead to motor and sensory dysfunction while differing in terms of lesion volume and extent of secondary damage. Transplantation of various NSCs indeed led to the generation of different cell fates correlating with the type of evoked injury.

Interestingly, rodent and human NSCs transplanted into a compression or a contusion lesion preferentially differentiated into oligodendrocytes [78,79,80,81,82,83,84,85,86,87,88,89,90,91] while cells transplanted into hemisected or transected lesions predominantly differentiated into astrocytes [92,93,94,95,96,97]. This effect seems to be independent of host species (mouse, rat), donor cell tissue and species (SGZ, SVZ, spinal cord, iPSCs from fibroblasts; mouse, rat, human) and age (adult, embryonic, fetal) of donor animals [81,89,94,98]—a relevant aspect for clinical translation. In the majority of the described cases, where these NSCs have been transplanted into a compression lesion, cells differentiated into oligodendrocytes (41–51%), followed by astrocytes (5–31.2%) and neurons (0–21%) [79,85,87,99]. Engrafted cells, which showed neither neuronal nor glial marker protein expression apparently remained as non-differentiated nestin-positive cells [82]; however, without developing signs of tumorigenesis. In contrast to this pro-oligodendroglial fate acquisition in compression lesions, NSCs transplanted in a hemisected or transected spinal cord model mostly differentiate into astrocytes, a few into oligodendroglial cells (except for up to 44% in one study) [97] and rarely into neurons as most studies state [95,96]. Therefore, the choice of the SCI model apparently exerts a great impact on the fate outcome, which could indeed influence the degree of cellular and functional regeneration. Nevertheless, there are always exceptions to the rule where environmental influences are not dominating [80,82,100,101]. Transplantation of fetal rat spinal cord NSCs into an adult rat SCI compression model resulted in a majority of GFAP-positive astrocytes (32.6%), a few 2′,3′-cyclic nucleotide 3′ phosphodiesterase- (CNP) positive oligodendrocytes (4.4%) and low degree of neuronal cells (5.9%) whereas precursor marker expression (such as neuronal Dcx or oligodendroglial NG2) was not investigated [80].

While most NSC transplantation based SCI studies focused on rodent models, Iwanami and colleagues used a primate contusion model. Upon cervical contusion in marmosets and subsequent transplantation of human NSCs, assessment of fate acquisition of the engrafted NSCs showed the following distribution: 46% GFAP-positive astrocytes, 25% nestin-positive stem cells, 21% β-III-Tub-positive neurons, and 5% Olig2-positive oligodendroglial cells [82]. Since in rodents a contused tissue environment generated overall higher oligodendroglial cell numbers as shown by the expression of myelin basic protein, remyelination events or axonal ensheathment by myelinating descendants of transplanted NSCs [79,84,85,90,102], these observations thus question how well rodent models can recapitulate the spinal cord pathology in higher mammals such as primates and consequently also in human.

Migration of transplanted NSCs towards an injury site, into lesion zones or close to demyelinated axonal fibers is key for a potential beneficial role of transplanted NSCs in terms of local neurotrophic factor secretion or remyelination. Considering the migratory potential of transplanted NSCs, it was found that migration exclusively occurs in spinal cords with a contusion or compression injury [83,84,85,86,87,89,91,99,103]. For hemisected and transected tissues the majority of studies did either not describe or discuss cell distribution or provided no explanations why migration was not observed [93,94,95,96,97,101,104]. Moreover, in most studies cells were grafted to positions rostral and/or caudal from the lesion sites and not directly into lesion zones thereby ensuring maximal survival rates of implanted cells [81,84,105]. Nevertheless, a single study directly assessed different positions of NSC injection and reported that a transplantation rostral and caudal of the lesion site leads to an increase in NSC survival when compared to injection into the lesion epicentre [86,87]. Such observations must therefore be considered in terms of clinical translation dealing with NSC-based regeneration therapies balancing survival of transplants vs. a limited degree of distribution which owning to the tissue sizes in higher mammals might indeed critically restrict a positive impact on cell replacement and functional restoration.

Despite the well described self-renewing potential of (neural) stem cells in vitro as well as in their discrete niches in vivo, transplanted stem cells showed little or no proliferation activities within host tissues in all SCI studies considered here [86,91,95,98]. Besides the described absence of tumorigenesis by NSCs in different SCI models, studies where NSCs were applied to an injury-free spinal cord also report no tumor formation [106,107]. The observed high differentiation rate and low proliferation potential of NSCs post-transplantation indicates that NSCs most likely sense and react to injury-specific cues which constitutes an important safety aspect. This is of even greater interest as indeed nearly all studies reported a certain degree of motor function recovery upon transplantation of diverse NSCs into the injured spinal cord independent of the chosen SCI model [82,102,103,105]. In this regard one has to take into account that studies with no observed functional benefit might remain unpublished, given that in SCI research recovery of lost functions constitutes a primary endpoint. However, reports on NSC populations failing to support neuroregeneration would still contribute to our still limited understanding of NSC biology under traumatic conditions.

The aforementioned studies revealing differences and similarities in different SCI models all used subacute or acute application conditions, in that NSC transplantation was performed immediately or shortly after injury. However, this does not necessarily reflect a patient’s situation who will first receive emergency surgery in order to stabilize general conditions and to avoid secondary damage preceding possible regeneration directed therapies. Moreover, for autologous stem cell transplantation a timely application of appropriate cell numbers is not feasible. Investigating chronic SCI models in which cells are transplanted several weeks after SCI might therefore be more relevant in terms of clinical translation and feasibility. In this regard, not only the type of induced injury (pressure or cut) appears to be important but also the disease process seems to account for the NSC fate. This particular question was addressed by several groups comparing chronic (several weeks after injury) to a subacute (days after injury) transplantation approaches. In two studies, a lower survival rate of NSCs transplanted after four and eight weeks into contusion lesions compared to a subacute transplantation process was observed [85,86]. While Karimi-Abdolrezaee and colleagues could not find any cells after one to two weeks following transplantation even with positive stimulating growth factors infused for seven days after transplantation, subsequent investigations [87,89,97] were able to detect engrafted cells several weeks later independent of the lesion model used (contusion, hemisection). Interestingly, even though NSCs were transplanted into chronic lesion conditions fate assessment analysis revealed that engrafted cells still differentiated mainly into glial cells and that only a few neurons were generated [86,97] bearing a high degree of similarity to the observations made in acute models. Thus, despite a lowered survival rate, the chronic lesion environment appears to still exert a beneficial influence on the differentiation of human NSCs and of iPSC-NPCs as two groups reported a more mature oligodendrocyte phenotype in stem cell derivatives upon transplantation into early chronic- or chronic lesions [98,99].

Of note, given that at least for transplanted OPCs heterogenic responses in gray vs. white matter have been described [108], a systematic investigation to reveal potential heterogeneity between gray/white matter instructed fates of transplanted NSCs has not been conducted yet. To our knowledge, a single study from 2008 reported that transplanted NSCs mainly accumulated in GM at the lesion site and with only a small percentage of cells found within WM regions [105]. As gray or white matter specific cues might indeed be relevant for survival or migration of transplanted NSCs such differences should be addressed more carefully when planning and analysing NSC transplantation studies in both, injury-free spinal cords and SCI models—also in light of the desired cellular outcome (neurons vs. oligodendroglia).

## 5. Conclusions

The high complexity of adult mammalian central nervous systems and the low degree of intrinsic repair capacity results in devastating functional impairments and persisting disabilities in most neuropathologies. The discovery of neural stem cells and their potential to give rise to all cell lineages of the CNS has revived the field in terms of functional cell replacement. Initial concerns related to exogenously applied NSCs and possible adverse effects were refuted as the majority of NSC transplantation studies revealed no tumor formation. Interestingly, studies investigating the outcome of NSC transplantations into healthy CNS tissue revealed dominant region-specific cues instructing the grafts. Despite a great heterogeneity among different CNS lesion models, transplanted NSC migration towards lesion sites and subsequent differentiation into to be replenished cell types are common observations in the majority of neuropathological models. These observations raise hope for the development of NSC-mediated CNS regeneration therapies as most neural stem cell types seem to be sensible to injury environments and their requirements and since even delayed NSC applications can confer functional benefits.

## Figures and Tables

**Table 1 ijms-20-00455-t001:** Summary of injury-free NSC transplantation studies.

Reference	Donor NSC Origin	Host Animal	CNS Region	Time Post-Transplantation	Outcome
Seidenfaden et al. 2006	SVZ Mice P5 or P75 -	Mice (C57BL/6) Six to ten weeks old	Striatum (mainly) Motor cortex Lateral posterior thalamic nucleus	3 wpt and 6 mpt	Survival: independent of the donor animal age Migration: no preference for GM or WM Cell fate: glial
Gage et al. 1995	Hpc Adult female Fischer 344 rats >3 months old 33 passages	Adult female Fischer 344 rats >3 months old	Hippocampus	1, 4, 8, 12 wpt	Survival: yes Migration: some up to 3 mm Cell fate: glial in Cc and neuronal in Hpc
Raedt et al. 2009	SVZ Male mice (C57BL/6×DBA2/J) - 10 passages	Sprague Dawley rats 175–200 g	Hippocampus	3 and 6 wpt	Survival: yes Migration: low degree Cell fate: astroglia (38.6%) and neuronal (5.8%)
Herrera et al. 1999	SVZ Mice (NSE-LacZ) 2–3 months old Directly isolated and transplanted	Male mice (CD-1) 2–3 months old	Cortex Striatum Hippocampus Olfactory bulb	2, 4, 6, 8 wpt	Survival: comparable between Str, Cx and OB, less survival in Hpc Migration: only in the OB Cell fate: non-neuronal type-C (or astrocyte) and type-A (neuronal precursor) phenotypes in Cx and Str, neurons in OB
Lois and Alvarez-Buylla 1994	SVZ Mice (NSE-LacZ) Adult -	Mice Adult	Lateral ventricle	30 dpt	Survival: - Migration: along RMS up to OB Cell fate: -
Fricker et al. 1999	Forebrain Human 6.5 to 9 weeks old 9 to 21 passages	Female Sprague Dawley rats Adult (250 g) Immunosuppressed	Dentate gyrus RMS Striatum Subventricular zone	2 and 6 wpt	Survival: yes Migration: low (DG, Str), high (SVZ, RMS) Cell fate: neuroblast features in SVZ and RMS neuronal in OB and Hpc, glial and neuronal in Str
Brock et al. 1998	SVZ Sprague Dawley rats P0, P1, or P2 Transplanted 24 h after isolation	Rats P0–P1	Subventricular zone	1 to 4 wpt	Survival: yes Migration: along RMS up to OB (SVZ-NSCs) cells remain at the injection site (VZ-NSCs) Cell fate: neurons
	VZ Rat embryos E16 to E17 Transplanted 24 h after isolation				

The table provides (from left-to-right) information on the original literature, donor cell origin (tissue, species, age, and passage number in culture if applicable), host animals (species, weight/age), region of transplantation within the CNS, time points of analysis, and features of transplanted cells (survival, migration, cell fate). A “-“ indicates that no information on this feature was given in the original publication. Abbreviations: SVZ—subventricular zone; P—postnatal day; dpt, wpt, mpt—days, weeks, months post-transplantation, respectively; GM/WM—gray- and white matter, respectively; Hpc—hippocampus; Cc—corpus callosum; Str—striatum; Cx—cortex; OB—olfactory bulb; RMS—rostral migratory stream; DG—dentate gyrus; VZ—ventricular zone.

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
