# Peer review of "Do Neural Stem Cells Have a Choice? Heterogenic Outcome of Cell Fate Acquisition in Different Injury Models"

_ijms, 2019, doi:10.3390/ijms20020455_

Round 1

Reviewer 1 Report

The review by Beyer et al.,  is an excellent and carefully prepared overview about transplantation studies of neural stem cells in healthy and injured brain. The mansucript is well-structured and esasy to read and understand. The discussed papers were well-chosen to summarize current knowledge.

I have only few particular points that could be amended

- line 74: should be "show" instead of "shows"

- line 108: the authors mention allotransplantation. Here it would be helpful to provide a short explanantion as this may not be know to every reader.

- line 131: no new paragraph since the authors are still discussing the same study. 

- line 175: I suggest to remove "brain and". In regard to what is discusses afterwards, it makes more sense to write" The astonishing especially RMS exerts on NSCs was corroborated...."

-  line 196: chapter 3 is not a real chapter, but more of an introduction into transplantations of NSC into injured/diseased brain. It therefore does not nescessarily need such a head line. Maybe subdivide the mansucript into 

1. healthy conditions

2. Brain pathology

            2.1 Dysmyelination neuropathies

            2.2 Traumatic brain injury...

- line 207: Dysmyelination neuropathies need a sentence to briefly explain the pathology and the shiverer mice, like it has been done for all following chapters. This chapter now jumps into methodology rightaway, assuming the reader to know about shivere mice etc.

-- line 282: TLE and Parkison's disease. This chapter nicely discusses TLE, but nothing (or so little that I overead it) is discussed about PD. If the authors want to discuss also PD, given the broad body of literatur on transplantation studies in PD, I also think it rather deverse a chapter of its own. 

- line 294: "... different fate choice compared to non-injured or TBI-injured hippocampi. Please specify to which conditions the following two sentence (294-298) refer to.

- line 319: NSC transplantation 7 months post-kainate injection revealed the majority of transplanted cells differentiated into astrocytes. Here the authors discussed that the microenvironment may have changed due to chronic neurodegeneration. While this is certainly true, another aspect that should not be forgotten is the fact that the hippocampal microenvironment (without any lesion) shifts towards a more gliogenic state with increasing age. From 6 months of age neurogenesis is decreasing steadily, and more astrocytes are generated. This aspect should ne included in the discussion of the results.

- line 635:  no new paragraph since the authors are still discussing the same study. 

- line 658: The sentence "Of note, to what degree...." is difficult to understand. Please rephrase.

Author Response

Thank you very much for your comments on our manuscript. The suggestions were very helpful in improving the manuscript and have all been implemented and highlighted in yellow.

Regarding your fifth suggestion regarding subdivision of the text we must state the following: Originally, we did not number the single chapters and this was then done post-submission. However, we now subdivided the main chapters as inspired by your suggestion.

Regarding your comment on TLE and Parkinson's disease: We agree with you and have now restricted ourselves to describing studies on the TLE paradigm.

Reviewer 2 Report

In this review, Beyer and colleagues, summarize, compare and contrast the outcomes of the existing neural stem cell transplantation approaches. The review is very well-written and will be great use to the community of neuroscientists who are interesting in regeneration after CNS injury. I have one minor comment. The first part of the review, which summarizes the injury-free NSC transplantation studies, can benefit from a table or a summary figure of some sort that highlights the ages of the transplanted cells/host, location and the outcome. Finally, please ensure that the origin of the transplanted NSCs (isolation/culturing method) for each study mentioned is clearly stated. 

Author Response

Thank you for your comments, they were appreciated. We absolutely agree with your first suggestion and therefore implemented Table 1. Additional clues and comments to Table 1 are highlighted in green. As to the second suggestion, which we think represents indeed a crucial point, we took again care in extracting as much as possible information on such details (additional information highlighted in green). However, since we already very carefully introduced each cell type used for transplantation in terms of derived species, niche, age, as well as (if provided) in vitro propagation prior to transplantation, when relevant (e.g. xenotransplantations), we also realized that unfortunately in many of the here discussed publications exactly this kind of information is often not provided to the desired extent.